# Use of *Haloxylon scoparium* Against Multidrug-Resistant Bacteria from Urinary Tract Infections

**DOI:** 10.3390/antibiotics14050471

**Published:** 2025-05-06

**Authors:** Fouad Bahri, Abdelhadi Boussena, Antoni Szumny, Youcef Bahri, El-Mokhtar Bahri, Adam Figiel, Piotr Juszczyk

**Affiliations:** 1Laboratory of Microbiology and Plant Biology, Faculty of Nature and Life Sciences, Abdelhamid Ibn Badis University, BP 188/227, Mostaganem 27000, Algeria; fouad.bahri@univ-mosta.dz (F.B.); abdelhadi.boussena@yahoo.fr (A.B.); youcefbahri99@gmail.com (Y.B.); mokhtar.bahri.91@gmail.com (E.-M.B.); 2Department of Food Chemistry and Biocatalysis, Faculty of Biotechnology and Food Sciences, Wrocław University of Environmental and Life Sciences, Norwida 31, 50-375 Wroclaw, Poland; 3Institute of Agricultural Engineering, Faculty of Life Sciences and Technology, Wrocław University of Environmental and Life Sciences, 51-630 Wrocław, Poland; adam.figiel@upwr.edu.pl; 4Department of Biotechnology and Food Microbiology, Faculty of Biotechnology and Food Science, Wroclaw University of Environmental and Life Sciences, 51-630 Wrocław, Poland; piotr.juszczyk@upwr.edu.pl

**Keywords:** bacteria MDR infections, 16S rRNA gene sequencing, natural substances, toxicity, HPLC-DAD

## Abstract

Background: The emergence of multidrug-resistant bacteria in the urinary tract and the decrease in the efficacy of antibiotics prompted us to evaluate the antibacterial activity of the methanolic extract of the aerial parts of *Haloxylon scoparium* against six isolated (MDR) bacteria. Methods: Phenolic compound profiling of the extract of interest was performed by HPLC-DAD. Acute oral toxicity was tested in vivo. The antibiotic susceptibility of the isolates was assessed against 23 antibiotics using the disk diffusion method. The identification of the isolates was performed by 16S rRNA gene sequencing. The antibacterial activity of the extract was assessed using agar well diffusion, minimum inhibitory concentrations (MICs), and minimum bactericidal concentrations (MBCs) methods. Results: Phenolic compound profiling of the extract revealed that epicatechin (85%) was the major compound. The extract also showed no symptoms of toxicity, adverse effects, or mortality in mice at the recommended dose. Overall, the extract at 200 µg/mL was effective against all isolates. The zones of inhibition ranged from 9.25 to 19.5 mm. Gram-positive *S. aureus* bacteria recorded the highest inhibitory effect with 19.5 mm against the five Gram-negative bacteria (9.25–17.25 mm). The MIC of the extracts against clinical isolates ranged from 50 to 100 µg/mL. The extract was bactericidal against *S. aureus*, *E. coli*, *E. ludwigii*, and *K. pneumoniae* with an MBC of 100, 100, 200, and 200 µg/mL, respectively. Conclusions: The results conclude that the extract could be an effective source of antimicrobial agents for the treatment of urinary tract infections caused by MDR bacteria.

## 1. Introduction

Urinary tract infections (UTIs) are one of the leading causes of morbidity and growing health care expenditure worldwide [1]. These are the most common bacterial infections seen in tertiary care hospitals, with higher morbidity and mortality among developing countries [2]. The global prevalence of urinary tract infections has led to increased antibiotic use. Antimicrobial resistance (MDR) is a rapidly emerging problem, particularly in developing countries, and urinary pathogens are among the most frequently resistant [3]. Today, multidrug-resistant (MDR) bacteria are increasingly prevalent, causing serious infections in healthcare establishments (nosocomial) as well as in the community. Bacteria have a broad genetic potential that enables them to transfer and acquire resistance to various classes of drugs that are currently in use [4]. In a recent publication, the WHO created a list of antibiotic-resistant bacteria “priority pathogen” belonging to 12 families that urgently require new antibiotics to combat them [5]. In both developed and developing countries, infections caused by MDR microorganisms lead to higher healthcare costs, longer hospital stays, increased morbidity, and higher mortality rates [6]. Diseases caused by drug-resistant pathogens are responsible for the deaths of at least 0.7 million people per year [7]. In African countries, treating infections caused by these pathogens is increasingly challenging due to limited effective therapies [8]. This ever-increasing burden is driving researchers to find new strategies. Natural products derived from plants are attracting the attention of researchers in the hope that they can eradicate antibiotic-resistant bacteria [9]. The long-term evolutionary process favors competition between pathogenic microorganisms and plants for survival. During this process, plants produce a wide range of bioactive secondary metabolites as a defense mechanism against infection by pathogenic microorganisms [10]. Some of these products offer new prospects for the development of antibiotics that are not only safe but also effective and inexpensive. It is essential to note that bacteria do not have the capacity to easily develop resistance to the phytochemicals present in plant extracts, which can be numerous and/or chemically complex [11,12]. Plants used in traditional Algerian medicine have been identified as potential sources of chemicals with valuable bioactive properties that can be used for medicinal applications, but there is little scientific evidence of their efficacy [13]. Plant antimicrobial screening is an important first step in the search for new antimicrobial drugs. The identification and evaluation of plants that possess antimicrobial properties is therefore of utmost importance. The Chenopodiaceae family is commonly utilized in traditional medicine and cultural practices [14]. It comprises 98 genera and around 1400 species. *Haloxylon scoparium* [=*Hammada scoparia* (Pomel) Iljin, *Arthrophytum scoparium* (Pomel) Iljin, *Salsola articulata* Cav, *Haloxylon articulatum* (Cav.)] [15,16] belongs to the Chenopodiaceae family. It is found worldwide, particularly in desert and semi-desert areas, and on soils containing high levels of salt. In Algeria, the plant grows spontaneously in the south and is known locally as “rimth”. The therapeutic qualities of this plant are widely recognized in traditional North African medicine, particularly for its effectiveness in treating inflammation, hepatitis, and cancer, and preventing obesity [17]. Traditionally, the infusion of the powdered aerial part of *H. scoparium* is used for its anti-diabetic, antiseptic, and anti-inflammatory effects [18]. The stems are used as a mordant to dye wool in traditional weaving [19]. Researchers have demonstrated that *H. scoparium* has antiproliferative and larvicidal activity [20]. In addition, the results of several studies have shown that *H. scoparium* extracts have an antimicrobial effect against a wide range of microorganisms [21]. This is due to their richness in phenolic compounds such as epicatechin, the major component of our extract.

Further evaluations are needed to obtain more relevant information on its antimicrobial activity. In the Algerian context, the antimicrobial activity of *H. scoparium* against clinical isolates has not been documented. In this regard, the current work was designed to evaluate the antibacterial properties of the methanolic extracts from *H*. *scoparium* aerial parts against MDR strains isolated from nosocomial infections.

## 2. Results

### 2.1. Polyphenol Compound Profile of Methanol Extract by HPLC-DAD

In this study, the characterization of phenolic compounds in the methanolic extract of the aerial part of *H. scoparium* was carried out using the HPLC-DAD method. Phenolic compounds were identified by comparing their peak retention time with that of the standard. The detected compounds were quantified by comparing the peak area of each compound with that of the standards. Data concerning retention time, Abs λ, and concentrations of identified polyphenolic compounds are summarized in Table 1, while Figure 1 represents a chromatogram of the phenolic compounds from the aerial part of *H. scoparium*.

### 2.2. Acute Oral Toxicity of Methanolic Extract

The acute oral toxicity study revealed that methanolic extract of the aerial part of *H. scoparium* did not induce symptoms of toxicity, adverse effects, or mortality 14 days after administration in any of the treated groups, suggesting that the median lethal dose of methanolic extract of the aerial part of *H. scoparium* is greater than 2000 mg.

### 2.3. Antibacterial Assays of H. scoparium Extracts

#### 2.3.1. Isolation of Bacterial Strains

Samples with bacterial concentrations ≥ 10^5^ cfu/mL on nutrient agar plates were treated as positive for urinary tract infection [22,23,24]. After subculture, three different bacterial strains were isolated on MacConkey agar plates (UTI-1, UTI-2, and UTI-3), one strain on nutrient medium (UTI-4), one strain on cetrimide medium (UTI-5), and one strain on Chapman medium (UTI-6). The colonies of the six isolates were as follows: circular, slimy, and domed for UTI-1, doughnut-shaped and salmon yellow in color for UTI-2, circular and salmon red in color for UTI-3, circular, smooth, and translucent white in color for UTI-4, mucous and yellow and green in color for UTI-5, and round, creamy, and pigmented (usually golden yellow) (UTI-6) (Figure 2). The bacterial strains showing visible growth were used further for analysis.

#### 2.3.2. Biochemical Investigation

The results of the Gram stain revealed that five isolated bacterial strains, namely UTI-1, UTI-2, UTI-3, UTI-4, and UTI-5, were Gram negative, except UTI-6, which was positive. The results of biochemical study are presented in Table 2.

#### 2.3.3. Isolation and Identification of Bacterial Isolates

A total of six MDR isolates were collected from various clinical samples from hospital patients in Chlef, Algeria. Based on 16S rRNA gene sequence, the isolates were identified as *K. pneumoniae* (UTI-1), *E. coli* (UTI-2), *E. hormaechei* (UTI-3), *E. ludwigii* (UTI-4), *P. aeruginosa* (UTI-5), and *S. aureus* (UTI-6). Nitrogen base sequences sorted from MDR isolates are presented as follows (Table 3):Isolate 1 (*Klebsiella pneumoniae*)GCTAACACATGCAAGTCGAGCGGTAGCACAGAGAGCTTGCTCTCGGGTGACGAGCGGCGGACGGGTGAGTAATGTCTGGGAAACTGCCTGATGGAGGGGGATAACTACTGGAAACGGTAGCTAATACCGCATAATGTCGCAAGACCAAAGTGGGGGACCTTCGGGCCTCATGCCATCAGATGTGCCCAGATGGGATTAGCTAGTAGGTGGGGTAACGGCTCACCTAGGCGACGATCCCTAGCTGGTCTGAGAGGATGACCAGCCACACTGGAACTGAGACACGGTCCAGACTCCTACGGGAGGCAGCAGTGGGGAATATTGCACAATGGGCGCAAGCCTGATGCAGCCATGCCGCGTGTGTGAAGAAGGCCTTCGGGTTGTAAAGCACTTTCAGCGGGGAGGAAGGCGTTGAGGTTAATAACCTTGGCGATTGACGTTACCCGCAGAAGAAGCACCGGCTAACTCCGTGCCAGCAGCCGCGGTAATACGGAGGGTGCAAGCGTTAATCGGAATTACTGGGCGTAAAGCGCACGCAGGCGGTCTGTCAAGTCGGATGTGAAATCCCCGGGCTCAACCTGGGAACTGCATTCGAAACTGGCAGGCTAGAGTCTTGTAGAGGGGGGTAGAATTCCAGGTGTAGCGGTGAAATGCGTAGAGATCTGGAGGAATACCGGTGGCGAAGGCGGCCCCCTGGACAAAGACTGACGCTCAGGTGCGAAAGCGTGGGGAGCAAACAGGATTAGATACCCTGGTAGTCCACGCCGTAAACGATGTCGATTTGGAGGTTGTGCCCTTGAGGCGTGGCTTCCGGAGCTAACGCGTTAAATCGACCGCCTGGGGAGTACGGCCGCAAGGTTAAAACTCAAATGAATTGACGGGGGCCCGCACAAGCGGTGGAGCATGTGGTTTAATTCGATGCAACGCGAAGAACCTTACCTGGTCTTGACATCCACAGAACTTTCCAGAGATGGATTGGTGCCTTCGGGAACTGTGAGACAGGTGCTGCATGGCTGTCGTCAGCTCGTGTTGTGAAATGTTGGGTTAAGTCCCGCAACGAGCGCAACCCTTATCCTTTGTTGCCAGCGGTTCGGCCGGGAACTCAAAGGAGACTGCCATGATAACTGGAGGAAGGTGGGGATGACGTCAGTCATCATGGCCCTTACGACAGGGCTACCACGTGCTACATGGCATTAC

These sequences were entered into the program DNA BLAST to show their types and how close they are to sequences in the Gene Bank; as the result of the analysis, these showed a similarity of (99%) between these sequences and the sequences of bacterial isolates registered in the Gene Bank with the number MT740433.1 (Figure 3).

b.Isolate 2 (*Escherichia coli*)GGCAGAAAGCTTGCTGTTTTTGCTGACGAGTGGCGGACGGGTGAGTAATGTCTGGGAATCTGCCTGATGGAGGGGGATAACTACTGGAAACGGTGGCTAATACCGCATAACGTCTCCGGACCAAAGAGGGGGATCTTCGGACCTCTTGCCATCGGATGAGCCCATATGGGATTAGCTAGTAGGTGGGGTAACGGCTCACCTAGGCGACGATCCCTAGCTGGTCTGAGAGGATGACCAGCCACACTGGAACTGAGACACGGCCCAGACTCCTACGGGAGGCAGCAGTGGGGAATATTGCACAATGGGCGCAAGCCTGATGCAGCCATGCCGCGTGTATGAAGAAGGCCTTCGGGTTGTAAAGTACTTTCAGCGGGGAGGAAGGGAATAAAGTTAATACCTTTGCTCATTGACGTTACCCGCAGAAGAAGCACCGGCTAACTCCGTGCCAGCAGCCGCGGTAATACGGAGGGTGCAAGCGTTAATCGGAATTACTGGGCGTAAAGCGCACGCAGGCGGTTTGTTAAGTCAGATGTGAAATCCCCGGGCTCAACCTGGGAACTGCATCTGATACTGGCTGGCTTGAGTCTCGTAGAGGGGGGTAGAATTCCATGTGTAGCGGTGAAATGCGTAGAGATCTGGAGGAATACCGGTGGCGAAGGCGGCCCCCTGGACAAAGACTGACGCTCAGGTGCGAAAGCGTGGGGAGCAAACAGGATTAGATACCCTGGTAGTCCACGCTGTAAACGATGTCGACTTGGAGGTTGTGCCCTTGAAGCGTGGCTTCCGGAGCTAACGCGTTAAGTCGACCGCCTGGGGAGTACGGCCGCAAGGTTAAAACTCAAATGAATTGACGGGGGCCCGCACAAGCGGTGGAGCATGTGGTTTAATTCGATGCAACGCGAAGAACCTTACCTGCTCTTGACATCCACCGAATTT

The result of the program DNA BLAST analysis showed a similarity of (98.28%) between sequences of bacterial isolates registered in the Gene Bank with the number ON653022.1 (Figure 4).

c.Isolate 3 (*Pseudomonas aeruginosa*)GGCAGCCTACACATGCAAGTCGAGCGGATGAAGGGAGCTTGCTCCTGGATTCAGCGGCGGACGGGTGAGTAATGCCTAGGAATCTGCCTGGTAGTGGGGGATAACGTCCGGAAACGGGCGCTAATACCGCATACGTCCTGAGGGAGAAAGTGGGGGATCTTCGGACCTCACGCTATCAGATGAGCCTAGGTCGGATTAGCTAGTTGGTGGGGTAAAGGCCTACCAAGGCGACGATCCGTAACTGGTCTGAGAGGATGATCAGTCACACTGGAACTGAGACACGGTCCAGACTCCTACGGGAGGCAGCAGTGGGGAATATTGGACAATGGGCGAAAGCCTGATCCAGCCATGCCGCGTGTGTGAAGAAGGTCTTCGGATTGTAAAGCACTTTAAGTTGGGAGGAAGGGCAGTAAGTTAATACCTTGCTGTTTTGACGTTACCAACAGAATAAGCACCGGCTAACTTCGTGCCAGCAGCCGCGGTAATACGAAGGGTGCAAGCGTTAATCGGAATTACTGGGCGTAAAGCGCGCGTAGGTGGTTCAGCAAGTTGGATGTGAAATCCCCGGGCTCAACCTGGGAACTGCATCCAAAACTACTGAGCTAGAGTACGGTAGAGGGTGGTGGAATTTCCTGTGTAGCGGTGAAATGCGTAGATATAGGAAGGAACACCAGTGGCGAAGGCGACCACCTGGACTGATACTGACACTGAGGTGCGAAAGCGTGGGGAGCAAACAGGATTAGATACCCTGGTAGTCCACGCCGTAAACGATGTCGACTAGCCGTTGGGATCCTTGAGATCTTAGTGGCGCAGCTAACGCGATAAGTCGACCGCCTGGGGAGTACGGCCGCAAGGTTAAAACTCAAATGAATTGACGGGGGCCCGCACAAGCGGTGGAGCATGTGGTTTAATTCGAAGCAACGCGAAGAACCTTACCTGGCCTTGACATGCTGAGAACTTTCCAGAGATGGATTGGTGCCTTCGGGAACTCAGACACAGGTGCTGCATGGCTGTCGTCAGCTCGTGTCGTGAGATGTTGGGTTAAGTCCCGTAACGAGCGCAACCCTTGTCCTTAGTTACCAGCACCTCGGGTGGGCACTCTAAGGAGACTGCCGGTGACAACCGGAGGAAGGTGGGGATGACGTCAAGTCATCTGGCCCTTACGGCAGGGCTACCACGTGCTACATGGTCGGTACAAGGGTTGCCAGCCGCGAG

The result of the program DNA BLAST analysis showed a similarity of (99%) between sequences of bacterial isolates registered in the Gene Bank with the number KJ819583.1 (Figure 5).

d.Isolate 4 (*Enterobacter hormaechei* subsp. *xiangfangensis*)ACAGGCAAGCAGCTTGCTGCTTCGCTGACGAGTGGCGGACGGGTGAGTAATGTCTGGGAAACTGCCTGATGGAGGGGGATAACTACTGGAAACGGTAGCTAATACCGCATAACGTCGCAAGACCAAAGAGGGGGACCTTCGGGCCTCTTGCCATCGGATGTGCCCAGATGGGATTAGCTAGTAGGTGGGGTAACGGCTCACCTAGGCGACGATCCCTAGCTGGTCTGAGAGGATGACCAGCCACACTGGAACTGAGACACGGTCCAGACTCCTACGGGAGGCAGCAGTGGGGAATATTGCACAATGGGCGCAAGCCTGATGCAGCCATGCCGCGTGTATGAAGAAGGCCTTCGGGTTGTAAAGTACTTTCAGCGGGGAGGAAGGCGATAAGGTTAATAACCTTGTCGATTGACGTTACCCGCAGAAGAAGCACCGGCTAACTCCGTGCCAGCAGCCGCGGTAATACGGAGGGTGCAAGCGTTAATCGGAATTACTGGGCGTAAAGCGCACGCAGGCGGTCTGTCAAGTCGGATGTGAAATCCCCGGGCTCAACCTGGGAACTGCATTCGAAACTGGCAGGCTAGAGTCTTGTAGAGGGGGGTAGAATTCCAGGTGTAGCGGTGAAATGCGTAGAGATCTGGAGGAATACCGGTGGCGAAGGCGGCCCCCTGGACAAAGACTGACGCTCAGGTGCGAAAGCGTGGGGAGCAAACAGGATTAGATACCCTGGTAGTCCACGCCGTAAACGATGTCGACTTGGAGGTTGTGCCCTTGAGGCGTGGCTTCCGGAGCTAACGCGTTAAGTCGACCGCCTGGGGAGTACGGCCGCAAGGTTAAAACTCAAATGAATTGACGGGGGCCCGCACAAGCGGTGGAGCATGTGGTTTAATTCGATGCAACGCGAAGAACCTTACCTACTCTTGACATCCAGAGAACTTACCAGAGATGCATTGGTGCCTTCGGGAACTCTGAGACAGGTGCTGCATGGCTGTCGTCAGCTCGTGTTGTGAAATGTTGGGTTAAGTCCCGCAACGAGCGCAACCCTTATCCTTTGTTGCCAGCGGTTCGGCCGGGAACTCAAAGGAAACTGCCAGTGATAACTGGAGGAAGGTGGGGATGACGTCAGTCATCTGGCCCTTACAATAGGGCTACCACGTGCTACATGGCGCAA

The result of the program DNA BLAST analysis showed a similarity of (99%) between sequences of bacterial isolates registered in the Gene Bank with the number MG923793.1 (Figure 6).

e.Isolate 5 (*Enterobacter ludwigii*)CTAACACATGCAAGTCGAACGGTAGCACAGAGAGCTTGCTCTCGGGTGACGAGTGGCGGACGGGTGAGTAATGTCTGGGAAACTGCCTGATGGAGGGGGATAACTACTGGAAACGGTAGCTAATACCGCATAACGTCGCAAGACCAAAGAGGGGGACCTTCGGGCCTCTTGCCATCAGATGTGCCCAGATGGGATTAGCTAGTAGGTGGGGTAACGGCTCACCTAGGCGACGATCCCTAGCTGGTCTGAGAGGATGACCAGCCACACTGGAACTGAGACACGGTCCAGACTCCTACGGGAGGCAGCAGTGGGGAATATTGCACAATGGGCGCAAGCCTGATGCAGCCATGCCGCGTGTATGAAGAAGGCCTTCGGGTTGTAAAGTACTTTCAGCGGGGAGGAAGGTGTTGTGGTTAATAACCGCAGCAATTGACGTTACCCGCAGAAGAAGCACCGGCTAACTCCGTGCCAGCAGCCGCGGTAATACGGAGGGTGCAAGCGTTAATCGGAATTACTGGGCGTAAAGCGCACGCAGGCGGTCTGTCAAGTCGGATGTGAAATCCCCGGGCTCAACCTGGGAACTGCATTCGAAACTGGCAGGCTAGAGTCTTGTAGAGGGGGGTAGAATTCCAGGTGTAGCGGTGAAATGCGTAGAGATCTGGAGGAATACCGGTGGCGAAGGCGGCCCCCTGGACAAAGACTGACGCTCAGGTGCGAAAGCGTGGGGAGCAAACAGGATTAGATACCCTGGTAGTCCACGCCGTAAACGATGTCGACTTGGAGGTTGTGCCCTTGAGGCGTGGCTTCCGGAGCTAACGCGTTAAGTCGACCGCCTGGGGAGTACGGCCGCAAGGTTAAAACTCAAATGAATTGACGGGGGCCCGCACAAGCGGTGGAGCATGTGGTTTAATTCGATGCAACGCGAAGAACCTTACCTACTCTTGACATCCAGAGAACTTTCCAGAGATGGATTGGTGCCTTCGGGAACTCTGAGACAGGTGCTGCATGGCTGTCGTCAGCTCGTGTTGTGAAATGTTGGGTTAAGTCCCGCAACGAGCGCAACCCTTATCCTTTGTTGCCAGCGGTCCGGCCGGGAACTCAAAGGAGACTGCCAGTGATAACTGGAGGAAGGTGGGGATGACGTCAGTCATCATGGCCCTTACGAGTAGGGCTACACACGTGCTACATGG

The result of the program DNA BLAST analysis showed a similarity of (99%) between sequences of bacterial isolates registered in the Gene Bank with the number OP986255.1 (Figure 7).

f.Isolate 6 (*Staphylococcus aureus*)CGCATGCTAATACATGCAAGTCGAGCGAACGGACGAGAAGCTTGCTTCTCTGATGTTAGCGGCGGACGGGTGAGTAACACGTGGATAACCTACCTATAAGACTGGGATAACTTCGGGAAACCGGAGCTAATACCGGATAATATTTTGAACCGCATGGTTCAAAAGTGAAAGACGGTCTTGCTGTCACTTATAGATGGATCCGCGCTGCATTAGCTAGTTGGTAAGGTAACGGCTTACCAAGGCAACGATGCATAGCCGACCTGAGAGGGTGATCGGCCACACTGGAACTGAGACACGGTCCAGACTCCTACGGGAGGCAGCAGTAGGGAATCTTCCGCAATGGGCGAAAGCCTGACGGAGCAACGCCGCGTGAGTGATGAAGGTCTTCGGATCGTAAAACTCTGTTATTAGGGAAGAACATATGTGTAAGTAACTGTGCACATCTTGACGGTACCTAATCAGAAAGCCACGGCTAACTACGTGCCAGCAGCCGCGGTAATACGTAGGTGGCAAGCGTTATCCGGAATTATTGGGCGTAAAGCGCGCGTAGGCGGTTTTTTAAGTCTGATGTGAAAGCCCACGGCTCAACCGTGGAGGGTCATTGGAAACTGGAAAACTTGAGTGCAGAAGAGGAAAGTGGAATTCCATGTGTAGCGGTGAAATGCGCAGAGATATGGAGGAACACCAGTGGCGAAGGCGACTTTCTGGTCTGTAACTGACGCTGATGTGCGAAAGCGTGGGGATCAAACAGGATTAGATACCCTGGTAGTCCACGCCGTAAACGATGAGTGCTAAGTGTTAGGGGGTTTCCGCCCCTTAGTGCTGCAGCTAACGCATTAAGCACTCCGCCTGGGGAGTACGACCGCAAGGTTGAAACTCAAAGGAATTGACGGGGACCCGCACAAGCGGTGGAGCATGTGGTTTAATTCGAAGCAACGCGAAGAACCTTACCAAATCTTGACATCCTTTGACAACTCTAGAGATAGAGCCTTCCCCTTCGGGGGACAAAGTGACAGGTGGTGCATGGTTGTCGTCAGCTCGTGTCGTGAGATGTTGGGTTAAGTCCCGCAACGAGCGCAACCCTTAAGCTTAGTTGCCATCATTAAGTTGGGCACTCTAAGTTGACTGCCGGTGACAAACCGGAAGAAAGGTGGGGATGACGTCAAATCATCATGCCCCTTATGATTTGGGCTACCC

The result of the program DNA BLAST analysis showed a similarity of (100%) between sequences of bacterial isolates registered in the Gene Bank with the number OP068064.1 (Figure 8).

#### 2.3.4. Antibiotic Susceptibility Test

Isolated bacteria were tested for their susceptibility profiles to antibiotics: penicillins, beta-lactams, cephems, carbapenems, aminoglycosides, tetracyclines, fluoroquinolones, quinolones, phenicols, nitrofurans, macrolides, ansamycins, and sulfonamides, according to CLSI guidelines [25]. CLSI consensus medical laboratory standards are the most widely recognized resources for continually improving the quality, safety, and effectiveness of testing. The results obtained are presented in Table 4. As an example, Figure 9 shows the results of E. ludwigii for antibiotic resistance (FOX, LVX, CIP, OFX, KAN, GMN, AMK, CMN, ERY, TET, QDA).

#### 2.3.5. Antibacterial Activity of Methanolic Extract of *Haloxylon scoparium*

The antibacterial activity of *H. scoparium* methanolic extracts against six isolates was determined by diffusion and microdilution methods of the well. The zones of inhibition are shown in Table 5 and the MIC and MBC results in Table 6.

This study demonstrates through statistical analysis (ANOVA) that the methanolic extract of *H. scoparium* has a dose-dependent effect on *E. ludwigii*. A significant increase in inhibition zones is observed with increasing concentration.

## 3. Discussion

The results of the characterization of phenolic compounds in the methanolic extract of the aerial part of *H. scoparium* revealed that the phenolic composition of this species is characterized by the presence of 18 compounds, 7 of which were phenolic acid derivatives (gallic acid, protocatchuic acid, chlorogenic acid, caffeic acid, vanillic acid, *p*-coumaric acid, and ferulic acid), and 10 were flavonoids such as flavan-3-ols (catechin (+) and epicatechin), flavones (apigenin and luteolin), flavonols (quercetin and rutin), flavone glycosides (luteolin glucoside and apigenin-7-glucoside), and flavanones (naringenin). The methanolic extract of *H. scoparium* showed the highest sum of the detected flavonoids (55.77 mg/g extract) due to its amount of epicatechin (47.516 mg/g extract), which represents more than 85% of the total amount of flavonoids detected. Other flavonoid compounds were less than 0.368 mg/g of extract, with the exception of catechin (3.208 mg/g of extract), luteolin glucoside (1.54 mg/g of extract), rutin (1.472 mg/g of extract), and apigenin-7-glucoside (1.272 mg/g of extract). Most phenolic acids such as ascorbic acid, ferulic acid, gallic acid, protocatchuic acid, vanillic acid, and p-coumaric acid were present in small amounts in the range of 0.0124–0.724 mg/g of extract, while chlorogenic acid (1.928 mg/g of extract) and caffeic acid (1.392 mg/g of extract) were detected in appreciable quantities. To the authors’ best knowledge, little research has been conducted on the phenolic composition of *H. scoparium* phenolic composition. The only reported studies were those of BenSalah et al. [19] and Jarraya et al. [26] on Tunisian *H. scoparium*. They reported the presence of isorhamnetin and 1-methylsalsolinol and a few triglycoside flavonols. In addition, Chao et al. [27] showed the presence of six phenolic compounds: cinnamic acid, coumaric acid, caffeoylquinic acid, catechol, cyanidin, and Chrysoeriol, which is considered to be a flavone. Recently, Benkherara et al. [28] reported that the methanol extract of the southeastern Algerian *H. scoparium* contained 29 phenolic compounds representing different classes, including phenolic acids and flavonoids, with a high content of gallic acid, followed by catechic acid and rutin. However, the presence of epicatechin at high levels was only detected in our study.

The LC-MS-MS profile of the hydroethanolic extract of Algerian *H. scoparium* revealed the presence of some phenolic compounds such as vanillin, naringenin, benzoic acid, myricetin, quercetin, rutin, caffeic acid, hydroxy4-coumarine, ascorbic acid, and gallic acid [29]. These differences in the phenolic composition of *H. scoparium* aerial part extracts could be explained by the impact of geographic and climatic conditions, as well as the polar or non-polar extraction solvent used [30,31]. Several studies have demonstrated that the majority of polyphenols have antimicrobial properties. Borges et al. [32] reported the antibacterial activity of gallic and ferulic acids. Protocatechuic and caffeic acids exhibited antibacterial activity against enterobacteria [33]. Chlorogenic acid also has antibacterial and antifungal effects by disrupting the structure of the cell membrane [34,35]. Various flavonoids such as catechin and its derivatives [36], luteolin derivatives [37], apigenin derivatives, and quercetin glycosides [38] have been identified to possess antibacterial properties. These compounds can act as antibiotics due to their ability to form complexes with soluble extracellular proteins, as well as with the cell walls of bacteria, which often leads to their inactivation and loss of function [39,40]. However, compounds present in the minority in the methanolic extract, such as ascorbic, ferulic, gallic, protocatchuic, vanillic and p-coumaric acid, naringinin, luteolin, and quercetin, could also contribute to the antibacterial activity of *H. scoparium* by synergy with the main compounds [41,42].

The acute oral toxicity study revealed that the median lethal dose of methanolic extract of the aerial part of *H. scoparium* is greater than 2000 mg. Lachkar et al. [43] and Kharchoufaetal [44]. in Morocco on *H. scoparium* showed that the acute toxicity study of the methanolic and aqueous extract causes mortality levels of 1/6 and 2/6, respectively, at a dose of 2000 mg/kg [43,44]. Therefore, the plant extract can be considered safe based on the Organization for Economic Co-operation and Development guideline, which recommends a maximum dose for acute toxicity of 2000 mg/kg.

A total of six different bacterial strains were isolated on MacConkey (UTI-1, UTI-2, and UTI-3), nutrient (UTI-4), cetrimide (UTI-5), and Chapman (UTI-6) agar plates. The biochemical characteristics of the strains studied are similar to those obtained by Kumar et al. [45] and Jackson et al. [46]. The isolates identified by 16S rRNA gene sequence are as follows: *K. pneumoniae* (UTI-1), *E. coli* (UTI-2), *E. hormaechei* (UTI-3), *E. ludwigii* (UTI-4), *P. aeruginosa* (UTI-5), and *S. aureus* (UTI-6). The detection of pathogenic strains and those resistant to antimicrobials used in the treatment of urinary tract infections is likely to help us understand the epidemiology of pathogens and could contribute to reducing the impact of these strains on human health. The multidrug-resistant bacteria isolated in our study have also been isolated by several authors [45,47,48].

Since antibiotics were discovered and introduced into medicine, bacteria have evolved diverse antibiotic resistance mechanisms [49,50]. These mechanisms are either the inhibition of cell wall synthesis or protein synthesis, or damage to the cytoplasmic membrane. This study showed that the antibiotic resistance of the tested isolates was high and alarming. This result is in line with several studies [51,52,53]. All bacteria showed resistance to antibiotics, but at variable levels, as shown in Table 4 and several other works [54,55]. The results in Table 4 show that *K. pneumoniae* and *P. aeruginosa* were resistant to eleven antibiotics, *E. coli* to eight antibiotics, *E. hormaechei* and *E. ludwigii* to seven antibiotics, and *S. aureus* to six antibiotics. It is noted that the five isolated Gram-negative bacteria are more resistant to antibiotics than the Gram-positive *S. aureus*. Any alteration in the outer membrane by Gram-negative bacteria, like changing the hydrophobic properties or mutations in porins and other factors, can create resistance. Gram-positive bacteria lack this important layer, which makes Gram-negative bacteria more resistant to antibiotics than Gram-positive ones [56,57]. Resistant bacteria could be the mutant form of common bacteria due to the non-judicial use of broad-spectrum antibiotics. Antibiotics are frequently prescribed in hospitals, are general practice, and are often administered before the pathogen’s culture and sensitivity results are known. Furthermore, genetic testing requires significant technical and financial resources, which may be lacking in many clinical laboratories [58,59,60]. As the distribution of causative organisms and antibiotic resistance rates vary according to time and place, the recent local data will be conducive to clinicians for choosing the best treatment [61,62,63]. As a result, not only will patients be treated with the correct antibiotics, but the misuse and overuse of antibiotics, which lead to rapid development and spread of resistance, will be minimized [63,64].

By increasing the concentration of the methanolic extract of *H. scoparium*, the inhibition diameters increase. At the concentration of 200 µg/mL, the inhibition zones vary from 9.25 to 19.5 mm. The highest diameter was recorded with S. aureus (19.5 mm), a Gram-positive bacterium. Gram-positive bacteria are more sensitive to antimicrobials than Gram-negative bacteria. Our results are identical to several works [56,57,65]. The inhibitory effect of the methanolic extract is due to phenolic compounds. The methanolic extract of *H. scoparium* is rich in flavonoids 55.77% and phenolic acids 8.766%. Flavonoids have been identified as polyphenolic compounds that are capable of exerting antibacterial activities by various mechanisms of action. According to several research studies, flavonoids can inhibit nucleic acid synthesis, cytoplasmic membrane function, and energy metabolism [64,66,67,68,69,70,71,72,73]. Plant-derived phenolic acids can inhibit the growth of many microorganisms [74,75,76,77,78]. Phenolic acids have membrane-active properties against bacteria, which cause leakage of cellular constituents, including nucleic acids, proteins, and inorganic ions such as potassium or phosphate. They act at both the membrane and cytoplasmic levels [78].

The MIC of the methanolic extract tested against selected MDR isolates ranged from 50 µg/mL to 200 µg/mL. *E. hormaechei*, *E. ludwigii*, *K. pneumoniae*, and *S. aureus* were susceptible to the methanolic extract of *H. scoparium* at a concentration of 50 µg/mL, while *P. aeruginosa* and *E. coli* were not susceptible. *H. scoparium* has previously been reported as antibacterial [8,21]. In terms of literature extraction, to date it has not been reported in the research literature that the methanol extract of the aerial parts of *H. scoparium* had an antibacterial effect against the clinical isolates. It also found that the bacteriostatic and bactericidal activity of the aerial part of *H. scoparium* was determined against MDR pathogens for the first time. When the results of the inhibition zone, bacteriostatic, and bactericidal concentration of both extracts of *H. scoparium* aerial part were compared, it became clear that the methanolic extract had better inhibitory activity against selected MDR isolates.

## 4. Materials and Methods

### 4.1. Plant Material Collection

The aerial parts of *H. scoparium* were collected in the city of Taghit from Bechar, Southwestern Algeria (altitude 676 m, latitude 31°17′08″ N, longitude 2°26′32″ W) in December 2023. It was identified with the aid of a botanist (Pr. Belhaçaini Hachimi, University of Sidi Bel-Abbès, Algeria), and a herbarium specimen (A.h. at ONA 2023) was archived at the microbiology and plant biology laboratory of the University of Mostaganem.

### 4.2. Extracts Preparation

The plant material was dried at room temperature for 30 days and then pulverized to a fine powder. The aerial parts of the plant (100 g) were macerated with 1 L of methanol and subjected to daily agitation (200 rpm) for 72 h at room temperature. The mixture was then filtered using Whatman No. 1 filter paper Sigma-Aldrich algiers, Algeria, and the filtrate was then concentrated at 40 °C in a R-100 Rotary Evaporator, Meiesseggstasse, Switzerland. A total of 16 g of the obtained extract was stored at 4 °C until further use [79].

### 4.3. HPLC-DAD of the Phenolic Profile of Methanol Extract

Phenolic compounds in the aerial part extract of *Haloxylon scoparium* methanol were analyzed using an HPLC-DAD system (Agilent Technologies 1200, Hampton, NH 03842, California, CA, USA). Separations were performed using a Zorbax Eclipse Plus C18 column (250 mm × 4.6 mm, particle size 5 µm., Agilent, California, CA, USA) at 40 ◦C with a flow rate of 0.5 mL/min, 5 µL of injection. Mobile phase A—0.1% formic acid, mobile phase B—acetonitrile was used as demonstrated in Table 7.

Standards and samples were detected using a Diode Array Detector (DAD) (UV–Vis) detector, in the range of 190–400 nm, Fulton, USA. The identification of phenolic compounds was carried out by comparison with the authentic standards at each retention time. The quantity of each phenolic compound identified was expressed in mg per gram of extract [80].

### 4.4. Acute Oral Toxicity Study in Mice

The experiment was carried out using the “up-and-down” test method on 12 healthy male Swiss albino mice weighing 28 ± 4 g and aged 8 to 10 weeks, according to OECD test guidelines no 425 [81]. The animals were randomly divided into four groups (n = 3 per group). Group 1 served as the control animals, and Groups 2, 3, and 4 as the drug-treated animals. Standard commercial food and water were provided. All mice were housed at 25 ± 2 °C in cages with 12 h day/night cycles and acclimatized to laboratory conditions for 2 weeks before starting the study. For experimental purposes, mice were fasted for 18 h with free access to water prior to the administration of the extract. Group 1 (control) animals received distilled water while Groups 2, 3, and 4 received the extract at concentrations of 500, 1000, and 2000 mg/kg, respectively. One hour after treatment, all mice were given food and water. The mice were carefully observed for any toxic effects during the first 6 h after the treatment period, and then every day for 14 days. The surviving mice were observed to determine the occurrence of toxic reactions.

### 4.5. Antibacterial Assays of H. scoparium Extracts

#### 4.5.1. Collection of Urine Samples

A total of 442 urine samples were collected from the Urology Department of Bedj Sisters Hospital in Chlef, Algeria (Table 8). The samples were collected over five months: August, September, October, November, and December 2023. Mid-stream urine samples were collected from male and female patients suspected of urinary tract infections. The ages ranged from 15 to 75 years. The samples were collected in 2 mL sterilized vials and immediately transported to the laboratory. The samples were used for immediate analysis or stored at 4 °C for later analysis.

#### 4.5.2. Isolation of Bacterial Species

The isolation of bacterial species was performed using the plate-spread technique. Urine samples were diluted 10^5^ times with sterile water, and 100 µL of the diluted sample was spread on nutrient agar plates. UTI-positive samples were identified, and morphologically different bacterial colonies were subcultured on MacConkey agar plates [82,83], nutrient agar [84], cetrimide agar [85], and Chapman agar [86]. The plates were incubated at 37 °C for 24 h, and plates that had no growth at the end of 24 h were incubated for an additional 24 h [82,86].

#### 4.5.3. Biochemical Investigation

Bacterial isolates were characterized by standard biochemical performances such as Gram stain, catalase production, oxidase production, methyl red, a Voges–Proskauer reaction, an indole test, starch hydrolysis, a citrate test, a mannitol test, and a nitrate test [82,87].

#### 4.5.4. Antibiotic Susceptibility Tests

An antibiotic susceptibility test of the six isolated bacteria was carried out using the diffusion method on Mueller–Hinton (MH) agar medium following the recommendations of the antibiogram committee of the French Society of Microbiology to determine the extent of resistance [88]. The antibiotics used were as follows: ampicillin (10 µg), amoxicillin/clavulanic acid (30 µg), cefazolin (30 µg), cefoxitin (30 µg), cefotaxime (30 µg), piperacillin (100 µg), ertapenem (10 µg), imipenem (10 µg), ciprofloxacin (5 µg), levofloxacin (5 µg), ofloxacin (5 µg), nalidixic acid (30 µg), amikacin (30 µg), gentamicin (10 µg), kanamycin (30 µg), tobramycin (10 µg), erythromycin (15 µg), clindamycin (2 µg), tetracycline (30 µg), trimethoprim/sulfamethoxazole (25 µg), quinupristin/dalfopristin (15 µg), rifampin (5 µg), and chloramphenicol (30 µg). Isolates classified as multidrug-resistant (MDR) are those that exhibit resistance to three distinct classes of antibiotics [89].

#### 4.5.5. DNA Isolation and PCR Amplification

Genetic sequencing was carried out to confirm the isolated strains [90]. Genomic DNA was isolated from freshly grown bacterial cultures using the DNA extraction kit (NucleoSpin from Macherey-Nagel, Düren, Germany), following the manufacturer’s instructions. Its quality was evaluated on 1.0% agarose gel, and a single band of large molecular weight DNA was observed. A single DNA fragment (approximately 1500 bp) was amplified on a thermal cycler (P100 from Biorad, California, CA, USA) using the universal primers 27F (5′-AGAGTTTGATCCTGGCTCAG-3′) and 1492R (5′-GGTTACCTTGTTACGACTT-3′) with the following condition: initial denaturation at 95 °C for 5 min; 30 cycles of denaturation at 95 °C for 30 s; annealing at 55 °C for 30 s; and 45 s for extension at 72 °C, with a final extension at 72 °C for 7 min. Amplified PCR products were revealed by 1.5% agarose gel electrophoresis. PCR amplicons were then purified using the Clean-up kit from Macherey-Nagel, Düren, Germany) following the protocol described by the supplier to remove any contaminants.

#### 4.5.6. Sequencing of 16S rRNA and Phylogenetic Analysis

Isolated and purified PCR products were sequenced using the Sanger technique [90] using an automated ABI 3100 sequencer, California, CA, USA, (Applied Biosystems) equipped with a BigDye Terminator v3.1 kit and the primers used for PCR amplification. The resulting sequences were analyzed and cleaned up using CHROMAS PRO software v. 2.1.8. The final sequences were then compared with those in the Gene Bank database using NCBI’s BLAST program (https://blast.ncbi.nlm.nih.gov/Blast.cgi, accessed on 31 October 2023) to identify the isolates studied on the basis of the percentage of homology with reference strains.

#### 4.5.7. Agar Well Diffusion Assay

To evaluate the antibacterial activity of the extract, the agar well diffusion method described by Perez et al. [91] and modified by Ahmad et Beg [92] was used. From frozen stock, fresh bacterial cultures were prepared and incubated overnight at 37 °C on Mueller–Hinton agar (MHA). After an incubation period of 18 to 24 h, a single colony of microorganisms was selected and inoculated into 5 mL of sterile saline (0.9% NaCl). Then, the turbidity was adjusted to a final density equivalent to the 0.5 McFarland standard, or 1.5 × 10^8^ CFU/mL. A total of 100 μL of each suspension was spread onto the surface of Mueller–Hinton agar (MHA), and 6 mm-diameter wells were aseptically perforated there using a sterile cork borer. Next, 15 μL of the test extract at concentrations of 50–200 µg/mL was deposited into the wells. Similarly, the extract was dissolved in 10% DMSO as negative controls, and 10 μg each of ertapenem and rifampicin were used as positive controls. After 2 h at 4 °C, all treated Petri dishes were incubated for 24 h at 37 °C. The diameter of the zone of inhibition (DZI) was measured and expressed in millimeters. The experiments were conducted three times.

#### 4.5.8. TTC Colorimetric Assay for the Evaluation of Minimum Inhibitory Concentration (MIC) and Minimum Bactericidal Concentration (MBC)

The MIC and MBC of the aerial parts extract were determined by the microdilution method in a 96-well microplate [93]. Briefly, a stock solution of the methanol extract (200 µg/mL) was prepared in 5% DMSO, and was then serially diluted to obtain test solutions of 200 to 3.125 µg/mL. Subsequently, 100 μL of each test solution was added to 95 μL of the Muller–Hinton broth in each well. Finally, 5 μL of each bacterial suspension (5 × 10^5^ CFU/mL) was added to each well. Positive growth control wells consisted of bacteria only in their appropriate medium. As a negative control, 5% DMSO was used. The plates were then incubated at 37 °C for 24 h. After incubation, 20 μL of 2 mg/mL 2,3,5-triphenyl tetrazolium chloride (TTC), dissolved in water, was applied to all wells as a growth indicator and then incubated for 30 min at 37 °C. Microorganism growth resulted in the appearance of a purple-red color, resulting from the reduction in TTC. The MIC was recorded as the lowest concentration of the extract that will inhibit the visible growth of bacteria after 24 h of incubation. The MBC was taken as the lowest extract concentration that showed no microbial growth [91,93]. Using a micropipette, 10 µL was taken from wells that did not show growth after incubation during MIC testing and streaked onto Mueller–Hinton agar. The plates were then incubated at 37 °C/24 h. All determinations were made in triplicate.

### 4.6. Data Analysis

The SPSS™ software (v. 26.0) was used for the statistical analysis of the data. The data were expressed as descriptive statistics in terms of relative frequency, and the mean ± standard deviations (SDs) of the triplicate measurements were calculated.

A one-way ANOVA was applied to compare the mean inhibition zones between the different concentrations. The hypotheses tested are as follows:H0: The means are equal across all concentrations.H1: At least one concentration has a significantly different mean.

Prior to conducting the ANOVA, its assumptions were evaluated:
Normality: The residuals appeared approximately normally distributed based on visual inspection.Homogeneity of variances: Variance across groups was roughly similar, satisfying the homoscedasticity assumption.

## 5. Conclusions

This study demonstrates, for the first time, the antibacterial activity of methanolic extracts of *Haloxylon scoparium* against the tested MDR clinical isolates. The methanolic extract showed no symptoms of toxicity with a better antibacterial effect and appears to be a potent antimicrobial agent that could be considered as a complementary and alternative drug for use against resistant pathogens. HPLC analysis showed in the methanolic extract of aerial parts of *H.scoparium* the presence of active inhibitors, including phenolic compounds that play a key role in the observed antibacterial properties of this extract. The most important compound in the extract is epicathrin (47.516 mg/g), followed by catchin (3.208 mg/g), luteolin (1.54 mg/g), rutin (1.472 mg/g), caffeic acid (1.392 mg/g), and apigenin-7-glucoside (1.272 mg/g) whose antibacterial effects have been documented in many studies.

The phenolic structure of the methanolic extract may play a role in the activity against the tested isolates, as the hydroxyl group in the structure allows these compounds to penetrate the cell and permeabilize the cytoplasmic membrane, leading to disruption of cellular metabolism. The results described are satisfactory given that bacterial resistance to antibiotics is becoming a global problem. More in vitro and in vivo studies are needed in a large number of clinical isolates to further investigate and standardize the inhibitory effects. This will be performed by identification, individual characterization of each bioactive compound, and an in-depth study on the action at the level of the different vital sites of the multidrug-resistant bacteria (DNA, Wall, Cytoplasmic Membrane, Ribosomes) for pharmaceutical use.

## Figures and Tables

**Figure 1 antibiotics-14-00471-f001:**
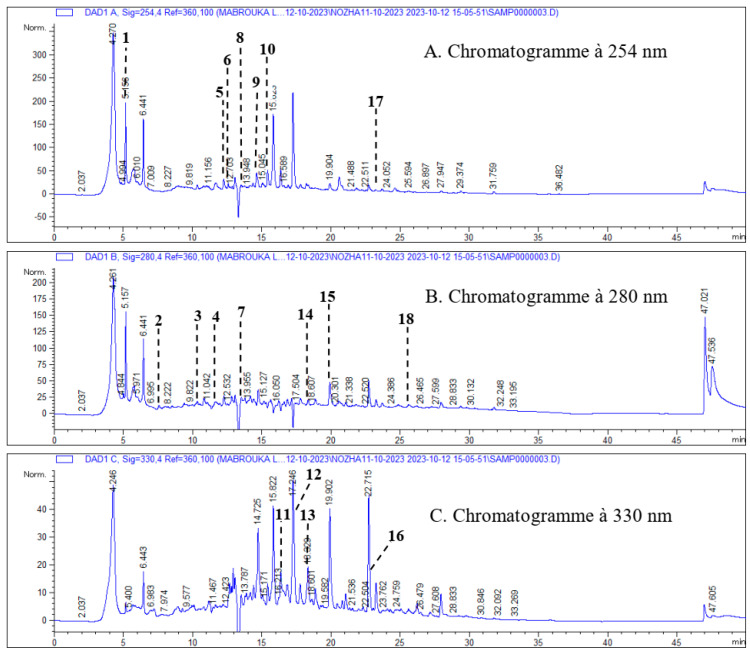
Chromatogram of phenolic compounds from the aerial part of *H. scoparium* recorded at 254 nm (**A**), 280 nm (**B**), and 330 nm (**C**); 1—ascorbic acid; 2—gallic acid; 3—3-4 dihydroxybenzoic acid; 4—chlorogenic acid; 5— (+) -catechin; 6—unknown; 7—caffeic acid; 8— (-) -epicatechin; 9—vanillic acid; 10—rutin; 11—luteolin-7-glucoside; 12- p-coumaric acid; 13—apigenin-7-glucoside; 14—ferulic acid; 15—naringenin 16- luteolin; 17—quercetin; 18—apigenin.

**Figure 2 antibiotics-14-00471-f002:**
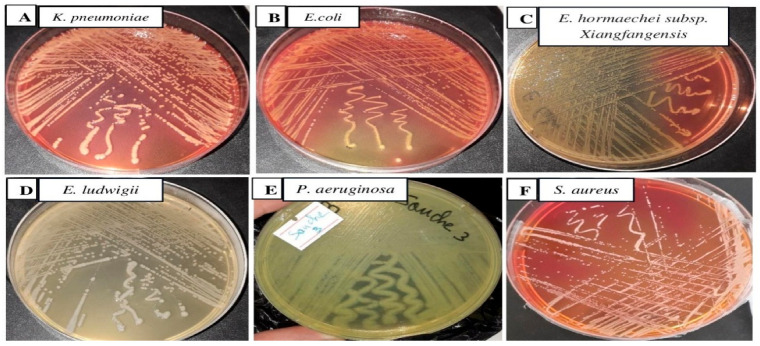
Bacteria isolated from hospitalized patients on MacConkey agar (**A**–**C**), nutrient agar (**D**), cetrimide (**E**), and Chapman (**F**) at 37 °C for 24 h.

**Figure 3 antibiotics-14-00471-f003:**
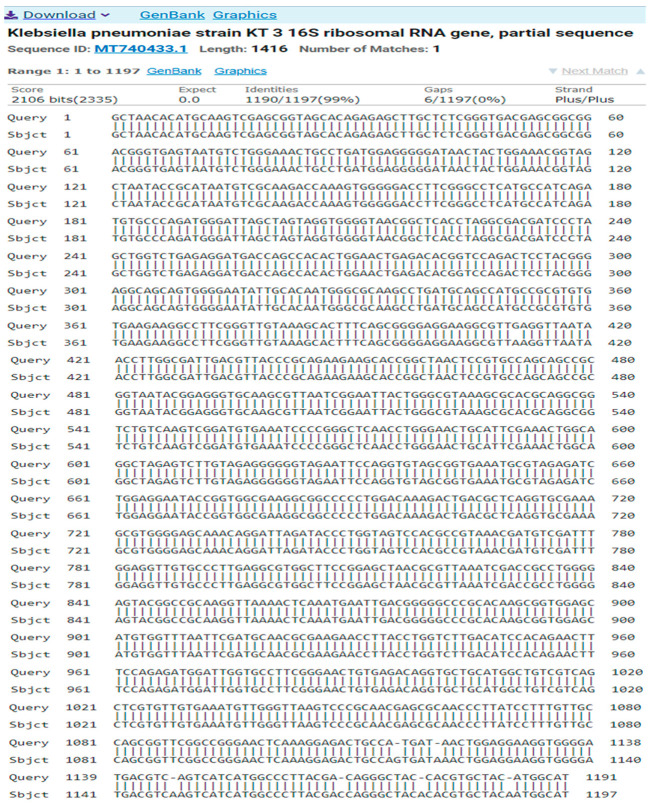
A comparison of sequences of the nitrogen base between the local isolate (*Klebsiella pneumoniae*) and the standard strain (MT740433.1).

**Figure 4 antibiotics-14-00471-f004:**
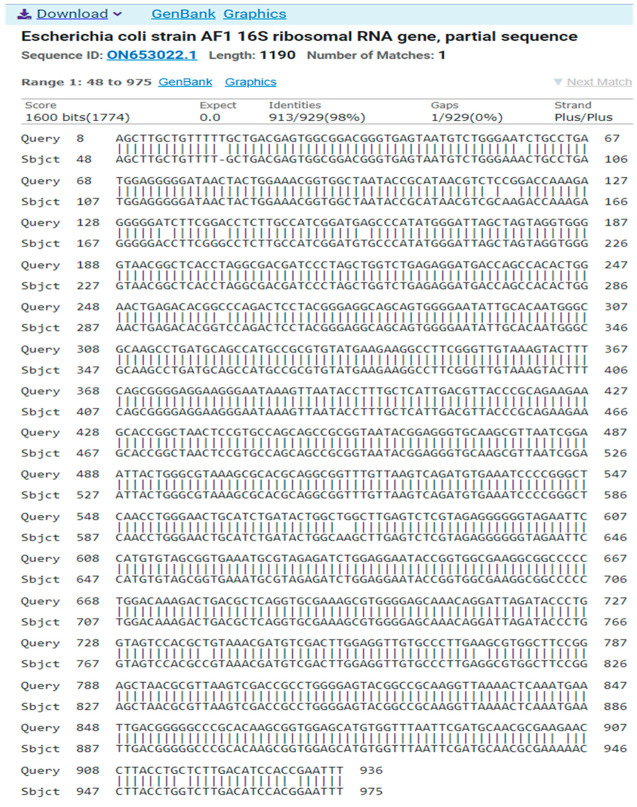
A comparison of sequences of the nitrogen base between the local isolate (*Escherichia coli*) and the standard strain (ON653022.1).

**Figure 5 antibiotics-14-00471-f005:**
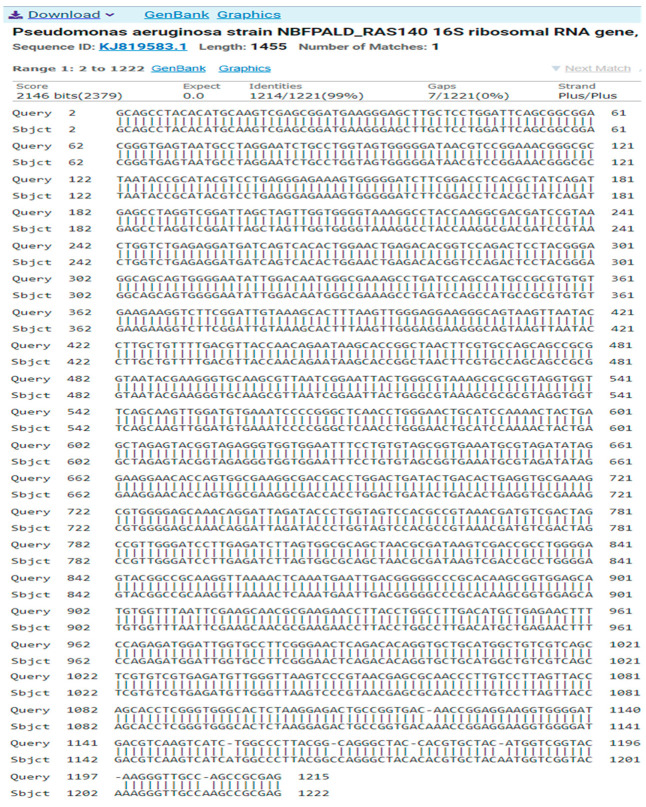
A comparison of sequences of the nitrogen base between the local isolate (*Pseudomonas aeruginosa*) and the standard strain (KJ819583).

**Figure 6 antibiotics-14-00471-f006:**
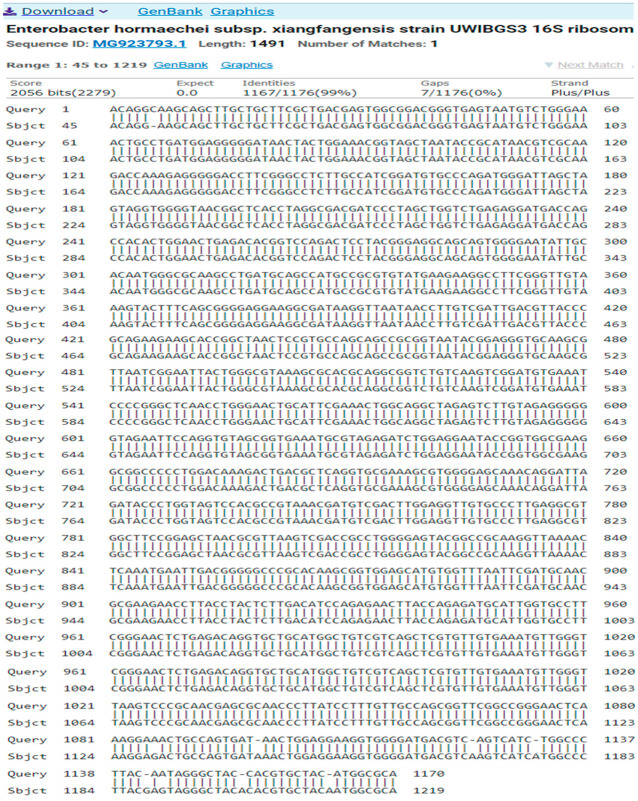
A comparison of sequences of the nitrogen base between the local isolate (*Enterobacter hormaechei* subsp. *Xiangfangensis*) and the standard strain (MG923793.1).

**Figure 7 antibiotics-14-00471-f007:**
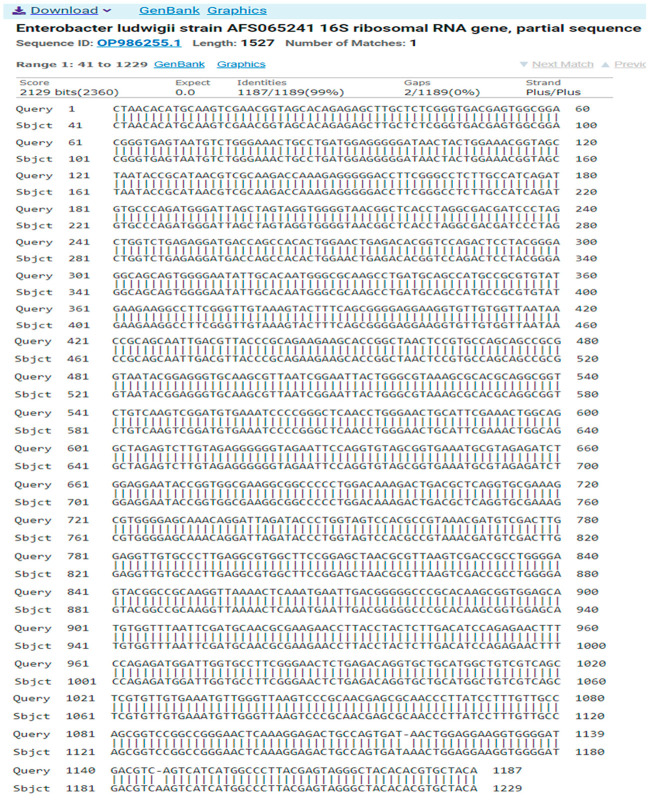
A comparison of sequences of the nitrogen base between the local isolate (*Enterobacter ludwigii*) and the standard strain (OP986255.1).

**Figure 8 antibiotics-14-00471-f008:**
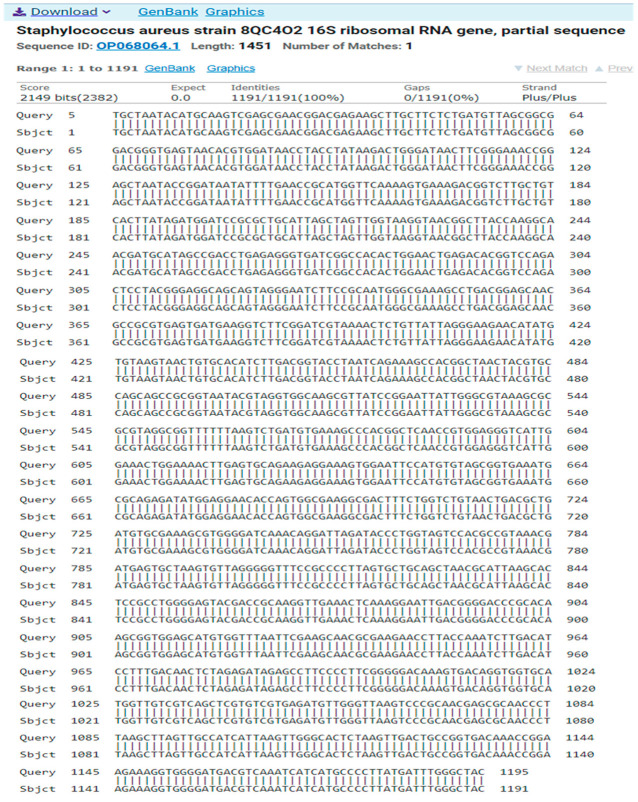
A comparison of sequences of the nitrogen base between the local isolate (*Staphylococcus aureus*) and the standard strain (OP068064.1).

**Figure 9 antibiotics-14-00471-f009:**
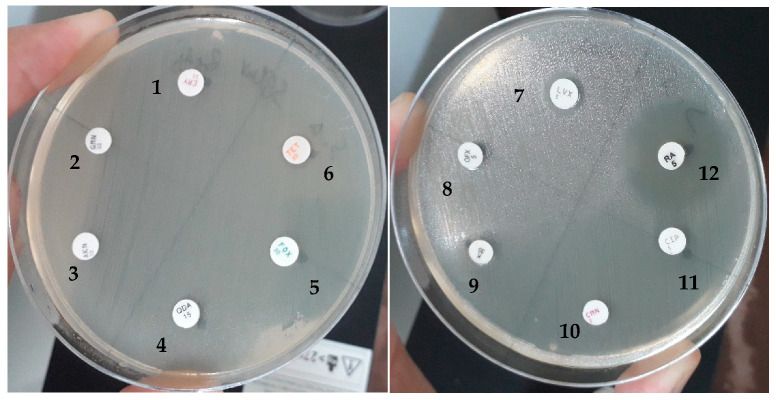
Antibiotic susceptibility of *Enterobactere ludwigi* isolates. 1. ERY (erythromycin); 2. GMN (gentamycin); 3. AMK (amikacin); 4. QDA (quinupristin/dalfopristin); 5. FOX (cefoxitin); 6. TET (tetracyclin); 7. LVX (levofloxacin); 8. OFX (ofloxacine); 9. KAN (kanamycin); 10. CMN (clindamycin); 11. CIP (ciprofloxacin); 12. RA (rifampicine).

**Table 1 antibiotics-14-00471-t001:** The retention time [Rt], absorbance [nm], and quantification of phenolic compounds in the methanolic extract of the aerial part of *H. scoparium*.

Peak No	Compound	Compound Families	Compound Subfamilies	Retention Time [min]	Concentration [mg/g Extract]	Abs λ [nm]
1	Ascorbic acid	Vitamins		4.771	0.0124	254
2	Gallic acid	Phenolic acids	Hydroxybenzoic acid	7.316	0.032	280
3	Protocatchuic acid			10.44	0.052	280
4	Chlorogenic acids		Hydroxycinnamic acid	11.44	1.928	280
5	Catechin	Flavonoids	Flavan-3-ol	12.16	3.208	254
6	Unknown	-	-	12.75	3.924	254
7	Caffeic acid	Phenolic	Hydroxycinnamic acid	13.505	1.392	280
8	Epicatechin	Flavonoid	Flavan-3-ol	13.521	47.516	254
9	Vanillic acid	Phenolic acids	Hydroxybenzoic acid	14.16	0.068	254
10	Rutin	Flavonoid	Flavonol glycoside	15.35	1.472	254
11	Luteolin glucoside	Flavonoid	Flavone glycosides	16.323	1.54	330
12	*p*-coumaric acid	Phenolic acids	Hydroxycinnamic acid derivative	17.138	0.724	330
13	Apigenin-7-glucoside	Flavonoid	Flavone glycosides	18.136	1.272	330
14	Ferulic acid	Phenolic acids	Hydroxycinnamic Acid derivative	18.339	0.184	280
15	Naringinin	Flavonoid	Flavanone	20.901	0.368	280
16	Luteolin	Flavonoid	Flavone	22.926	0.096	330
17	Quercetin	Phenolic	Flavonol	23.048	0.252	254
18	Apigenine	Flavonoid	Flavone	25.704	0.216	280
Sum of all compounds					64.536	
Sum of flavonoids					55.77	
Sum of phenolic acids					8.766	

**Table 2 antibiotics-14-00471-t002:** Biochemical investigation reports of isolated germs.

Biochemical Test	UTI-1	UTI-2	UTI-3	UTI-4	UTI-5	UTI-6
Gram staining	-	-	-	-	-	+
Motility	Non-motile	Motile	Motile	Motile	Motile	Non-motile
Oxidase	-	-	-	-	+	-
Catalase	+	+	+	+	+	+
Nitrate	+	+	+	+	+	+
Methyl red	+	+	+	+	-	+
Voges–Proskaeur	+	-	+	+	-	+
Citrate	+	-	+	+	+	+
Indole	-	+	-	-	-	-
Starch hydrolysis	-	-	+	+	-	+

-: négative; +: positive.

**Table 3 antibiotics-14-00471-t003:** Sequences producing significant alignments.

Description	Scientific Name	Max Score	Totale Score	Query Cover	E Value	Per. Ident	Acc. Len	Accession
Klebsiella pneumoniae strain KT 3 16S ribosomal RNA gene, partial sequence	*Klebsiella pneumoniae*	2167	2167	100%	0.0	99.33%	1416	MT740433.1
Escherichia coli strain AF1 16S ribosomal RNA gene, partial sequence	*Escherichia coli*	1626	1626	99%	0.0	98.28%	1190	ON653022.1
Pseudomonas aeruginosa strain NBFPALD_RAS140 16S ribosomal RNA gene, partial sequence	*Pseudomonas aeruginosa*	2209	2209	100%	0.0	99.43%	1455	KJ819583.1
Enterobacter hormaechei subsp. xiangfangensis strain UWIBGS3 16S ribosomal RNA gene, partial sequence	*Enterobacter hormaechei* subsp. *xiangfangensis*	2115	2115	100%	0.0	99.23%	1491	MG923793.1
Enterobacter ludwigii strain AFS065241 16S ribosomal RNA gene, partial sequence	*Enterobacter ludwigii*	2183	2183	100%	0.0	99.75%	1527	OP986255.1
Staphylococcus aureus strain 8QC4O2 16S ribosomal RNA gene, partial sequence	*Staphylococcus aureus*	2200	2200	99%	0.0	100%	1451	OP068064.1

**Table 4 antibiotics-14-00471-t004:** Panel of MDR clinical bacterial isolates used in this study, as well as their origin and resistance profiles.

Clinical MDR Isolate	Antibiotic Resistance Patterns
*K. pneumoniae*	AMP, AMC, CZN, FOX, CTX, ETP, AMK, GMN, NAL, TET, SXT
*E. coli*	AMP, AMC, FOX, CTX, ETP, GMN, NAL, SXT
*P. aeruginosa*	PI, LVX, CIP, GMN, TOB, IMP
*E. hormaechei*	AMP, AMC, CZN, CTX, GMN, SXT
*E. ludwigii*	FOX, LVX, CIP, OFX, KAN, GMN, AMK, CMN, ERY, TET, QDA
*S. aureus*	FOX, KAN, GMN, AMK, CMN, ERY, TET, QDA

AMP (ampicillin), AMC (amoxicilline/clavulanic acid), CZN (cefazolin), FOX (cefoxitin), CTX (cefotaxime), PI (piperacillin), ETP (ertapenem), IMP (imipenem), CIP (ciprofloxacin), LVX (levofloxacin), OFX (ofloxacine), NAL (nalidixic acid), AMK (amikacin), GMN (gentamycin), KAN (kanamycin), TOB (tobramycin), ERY (erythromycin), CMN (clindamycin), TET (tetracyclin), SXT (trimethoprim/sulfamethoxazole), QDA (quinupristin/dalfopristin).

**Table 5 antibiotics-14-00471-t005:** Antibacterial activity of *H. scoparium* (aerial parts) against 6 MDR isolates.

	The Diameter of Inhibition Zones (mm) (mean ± SD)						
MDR Bacteria	Methanolic Extract (µg/mL)				Positive Control		Negative Control
	50	100	150	200	ETP (10 μg)	RP (10 μg)	DMSO (10%)
*K. pneumoniae*	8.0 ± 0.80	9.0 ± 0.71	11.0 ± 0.4	14.75 ± 0.35	14.5 ± 1.2	-	0
*E. hormaechei*	7.0 ± 0.00	7.37 ± 0.13	9.5 ± 0.17	11.0 ± 0.35	30.50 ± 0.2	-	0
*E. ludwigii*	8.75 ± 0.0	11.5 ± 0.5	14.5 ± 0.5	16.0 ± 0.5	-	22.50 ± 0.8	0
*E. coli*	0	8.0 ±1.41	14.0 ±0.23	17.25 ± 0.35	10.0± 0.8	-	0
*P. aeruginosa*	0	7.5 ± 0.71	8.5 ± 0.43	9.25 ± 0.32	30.50 ± 0.2	-	0
*S. aureus*	7.5 ± 0.00	14.25 ± 1.0	17.5 ± 0.16	19.5 ± 0.71	-	21.0 ± 0.5	0

±: standard error of given value; -: not tested; ETP: ertapenem; RP: rifampicin; DMSO: dimethyl sulfoxide.

**Table 6 antibiotics-14-00471-t006:** MIC and MBC values of methanolic extract of aerial parts of *H. scoparium* against 6 MDR pathogens.

MDR Bacteria	Methanolic Extract (µg/mL)		
	MIC	MBC	MBC/MIC
*K. pneumoniae*	100	200	4
*E. hormaechei*	100	>200	ND
*E. ludwigii*	100	200	4
*E. coli*	50	100	2
*P. aeruginosa*	50	>200	ND
*S. aureus*	50	100	2

>200 shows concentration greater than 200; ND: not defined.

**Table 7 antibiotics-14-00471-t007:** HPLC-DAD gradient solvent system for phenolic compound separation.

Time/min	Solvent A (%)	Solvent B (%)
0–22	90	10
22–32	50	50
32–40	100	0
40–44	100	0
44–50	10	90
50	10	90

**Table 8 antibiotics-14-00471-t008:** The distribution of the 442 samples according to age and sex.

Age (Years)	Female	Male
15–24	32	70
25–45	57	80

## Data Availability

Data are contained within the article.

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
