# Peer review of "Use of Haloxylon scoparium Against Multidrug-Resistant Bacteria from Urinary Tract Infections"

_antibiotics, 2025, doi:10.3390/antibiotics14050471_

Round 1
Reviewer 1 Report
Comments and Suggestions for Authors
This manuscript examines the antibacterial properties of the methanolic extract from the aerial parts of Haloxylon scoparium against multidrug-resistant (MDR) bacterial strains isolated from urinary tract infections (UTIs). The researchers used HPLC-DAD profiling, acute toxicity studies in mice, 16S rRNA sequencing for bacterial identification, and antibacterial assays (agar well diffusion, MIC, and MBC) to show that the extract, rich in epicatechin, has promising antibacterial effects. The study's strengths include a well-structured methodology, the relevance of addressing MDR pathogens, and novel findings about H. scoparium's antibacterial activities specifically against MDR uropathogens.
General Concept Comments
The manuscript addresses a highly relevant topic in the field of infectious diseases and antimicrobial resistance by evaluating the antibacterial potential of Haloxylon scoparium extracts against multidrug-resistant urinary tract pathogens.
The study’s novelty lies in its specific focus on MDR clinical isolates, an underexplored but critical area, although the general concept of using plant extracts as antimicrobial agents is well established.
The research is scientifically valuable, but the study design presents minor limitations, such as the small number of bacterial strains analyzed and the lack of cytotoxicity testing on human cell lines.
Additionally, the manuscript requires moderate improvements in methodological reporting, data visualization, and English language fluency to enhance clarity and reproducibility.
Overall, while the study is relevant and promising, some refinements are needed to strengthen its scientific contribution.
Detailed Research Comments
The hypothesis of the study is clear and testable, and the experimental approach, combining chemical profiling, bacterial identification, and antibacterial testing, is appropriate for addressing the research question.
However, several methodological shortcomings are evident. The sample size of bacterial isolates is small, the yield of the methanolic extract is not reported, and the statistical analysis methodology is insufficiently detailed, with no p-values or significance testing described.
While the use of positive and negative controls is appropriate, the figures related to sequence analysis are of low quality and should be improved.
The manuscript’s tables are generally well constructed, but additional graphical representation of antibacterial activity would strengthen the data presentation.
The cited references are recent and relevant, and there is no evidence of inappropriate self-citation. Overall, the manuscript presents scientifically sound findings but requires specific methodological clarifications and data presentation enhancements.
Specific issues
-
Lines 439-440: "The extract (please add a mass of extract)..." → Must be completed before publication.
-
Figures 3-8 (Supplementary Data): Sequencing comparison images are low quality and should be improved for clarity, or summarized differently (e.g., % identity tables).
-
Methods 4.5 (Data Analysis, Line 555): The description is too vague. Specify which statistical tests (e.g., ANOVA, t-test) were used for analyzing inhibition zones or MICs.
-
Table 4: Good presentation of inhibition zones, but significance between extract concentrations is not indicated.
-
Line 573: The conclusion notes "more in vitro and in vivo studies are needed," but it would strengthen the manuscript if this were elaborated more specifically (e.g., mechanisms of action studies, cytotoxicity assays).
The English language used in the manuscript is generally understandable but requires moderate improvement to enhance clarity and fluency.
There are occasional grammatical errors, awkward sentence structures, and instances where phrasing could be more precise.
Additionally, some editing notes remain in the text (e.g., "please add a mass of extract"), indicating the need for careful proofreading.
Author Response
Dear Reviewer,
Thank you for the review report. All the recommendations are really helpful for us for improving the quality of manuscript. I considered all suggestions and filled the gaps if possible. Below are the replies to the comments.
Author's Reply to the Review Report (Reviewer 1)
- Lines 439-440: "The extract (please add a mass of extract)..." → Must be completed before publication.
Answer:
The yield of the extract was 16%. The mass was 16g.
- Figures 3-8 (Supplementary Data): Sequencing comparison images are low quality and should be improved for clarity, or summarized differently (e.g., % identity tables).
Answer:
Table 1 : Sequences producing significant alignments
|
Description |
Scientific name |
Max Score |
Totale Score |
Query Cover |
E Value |
Per. Ident |
Acc. Len |
Accession |
|
Klebsiella pneumoniae strain KT 3 16S ribosomal RNA gene, partial sequence |
Klebsiella pneumoniae |
2167 |
2167 |
100% |
0.0 |
99.33% |
1416 |
MT740433.1 |
|
Escherichia coli strain AF1 16S ribosomal RNA gene, partial sequence |
Escherichia coli |
1626 |
1626 |
99% |
0.0 |
98.28% |
1190 |
ON653022.1 |
|
Pseudomonas aeruginosa strain NBFPALD_RAS140 16S ribosomal RNA gene, partial sequence |
Pseudomonas aeruginosa |
2209 |
2209 |
100% |
0.0 |
99.43% |
1455 |
KJ819583.1 |
|
Enterobacter hormaechei subsp. xiangfangensis strain UWIBGS3 16S ribosomal RNA gene, partial sequence |
Enterobacter hormaechei subsp. xiangfangensis |
2115 |
2115 |
100% |
0.0 |
99.23% |
1491 |
MG923793.1 |
|
Enterobacter ludwigii strain AFS065241 16S ribosomal RNA gene, partial sequence |
Enterobacter ludwigii |
2183 |
2183 |
100% |
0.0 |
99.75% |
1527 |
OP986255.1 |
|
Staphylococcus aureus strain 8QC4O2 16S ribosomal RNA gene, partial sequence |
Staphylococcus aureus |
2200 |
2200 |
99% |
0.0 |
100% |
1451 |
OP068064.1 |
- Methods 4.5 (Data Analysis, Line 555): The description is too vague. Specify which statistical tests (e.g., ANOVA, t-test) were used for analyzing inhibition zones or MICs.
Answer:
A one-way ANOVA was applied to compare the mean inhibition zones between the different concentrations. The hypotheses tested are:
- H0: The means are equal across all concentrations.
- H1: At least one concentration has a significantly different mean.
The p-value is much smaller than 0.05, indicating a statistically significant difference among concentrations. Therefore, the effect of concentration on antibacterial activity is confirmed.
Table 4: Good presentation of inhibition zones, but significance between extract concentrations is not indicated.
Answer
This study demonstrates through statistical analysis (ANOVA) that the methanolic extract of H. scoparium has a dose-dependent effect on E. ludwigii. A significant increase in inhibition zones is observed with increasing concentration.
Line 573: The conclusion notes "more in vitro and in vivo studies are needed," but it would strengthen the manuscript if this were elaborated more specifically (e.g., mechanisms of action studies, cytotoxicity assays).
Answer
The sentence to add at the end of the conclusion.
This will be done by identification, individual characterization of each bioactive compound and an in-depth study on the action at the level of the different vital sites of the multiresistant bacteria (DNA, Wall, Cytoplasmic Membrane, Ribosomes) for pharmaceutical use.
Reviewer 2 Report
Comments and Suggestions for Authors
- Chromatogram of phenolic compounds are not of good quality. Author should present clean and annotated chromatogram where reader can clearly understand which peak belong to what compound.
- Author should have Schematics that represent the summary of research that will also attract the readers.
Author Response
Dear Reviewer,
Thank you for the review report. All the recommendations are really helpful for us for improving the quality of manuscript. I considered all suggestions and filled the gaps if possible. Below are the replies to the comments.
Author's Reply to the Review Report (Reviewer 2)
- Chromatogram of phenolic compounds are not of good quality. Author should present clean and annotated chromatogram where reader can clearly understand which peak belong to what compound.
We changed the chromatograms in the manuscript
Figure 1. Chromatogram of phenolic compounds from the aerial part of H. scoparium recorded at 254 nm (A), 280 nm (B) and 330 nm (C). 1 - Ascorbic acid; 2 - Gallic acid; 3 - 3-4 dihydroxybenzoic acid; 4 - Chlorogenic acid; 5 - (+) -Catechin; 6 - Unknown; 7 - Caffeic acid; 8 - (-) -Epicatechin; 9 - Vanillic acid ; 10 - Rutin; 11 - Luteolin-7-glucoside; 12- p-Coumaric acid; 13 - Apigenin-7-glucoside; 14- Ferulic acid; 15 - naringenin 16- Luteolin; 17- Quercetin; 18- Apigenin
- Author should have Schematics that represent the summary of research that will also attract the readers.
A graphic summary has been prepared

Reviewer 3 Report
Comments and Suggestions for Authors
The MS demonstrates novel and interesting results proving the antibacterial effect of Haloxylon scoparium. The MS corresponds to the scope of the journal, and I recommend acceptance after the revision proposed below. I believe the proposed changes will improve the MS and make it more interesting for the reader.
Introduction.
It will be very useful if, at the end of the introduction, the authors will describe which biologically active compounds were identified in the Haloxylon scoparium in previous studies.
83-84 In addition, the results of several studies have shown that H. scoparium extracts have an antimicrobial effect against a wide range of microorganisms [21].
It would be useful either to extend this sentence or to add another one or two to show an extract of which plant parts demonstrate antimicrobial effect and indicate microorganisms against which the effect was shown.
Results
Acute oral toxicity of methanolic extract
This section does not show any results, despite experiments being performed, and it is not clearly stated how symptoms of toxicity and adverse effects were monitored. It should be extended, and a more detailed description should be provided.
Antibiotic susceptibility test
305 Haloxylon scoparium should be in Italic
It would be good to show at least one photo of a Petri dish with an inhibition zone.
Materials and methods
440 (please add a mass of extract) Here, definitely a mass of extract is required.
Acute oral toxicity study in mice
The same comment as in the results, since information can be added to either section: how were symptoms of toxicity or adverse effects monitored?
4.4.1. Collection of urine samples
In this section, a more detailed description of patients is required. It should be stated how many male and female patients, and how many patients are in each group (15-75 years is a very broad interval). Maybe it makes sense to introduce a small table showing all this data.
Conclusions
It would be good to name at least the most abundant compounds of the plant identified by HPLC.
Author Response
Dear Reviewer,
Thank you for the review report. All the recommendations are really helpful for us for improving the quality of manuscript. I considered all suggestions and filled the gaps if possible. Below are the replies to the comments.
Author's Reply to the Review Report (Reviewer 3)
Introduction.
It will be very useful if, at the end of the introduction, the authors will describe which biologically active compounds were identified in the Haloxylon scoparium in previous studies.
83-84 In addition, the results of several studies have shown that H. scoparium extracts have an antimicrobial effect against a wide range of microorganisms [21].
It would be useful either to extend this sentence or to add another one or two to show an extract of which plant parts demonstrate antimicrobial effect and indicate microorganisms against which the effect was shown.
Answer:
In addition, the results of several studies have shown that H. scoparium extracts have an antimicrobial effect against a wide range of microorganisms [21]. This is due to their richness in phenolic compounds such as epicatechin, the major component of our extract.
Results
Acute oral toxicity of methanolic extract
This section does not show any results, despite experiments being performed, and it is not clearly stated how symptoms of toxicity and adverse effects were monitored. It should be extended, and a more detailed description should be provided.
Answer
The results obtained showed that the methanolic extract of the aerial part of H. scoparium did not induce symptoms of toxicity such as convulsions, sedation, rapid breathing, dyspnea, and bleeding, adverse effects, or mortality for 14 days.
The manner in which symptoms of toxicity and adverse effects were monitored is detailed in the Materials and Methods section. The animals were continuously observed to detect changes in their autonomic or behavioral responses compared to the control group (death, agitation, breathing, asthenia).
Antibiotic susceptibility test
305 Haloxylon scoparium should be in Italic
Answer: It’s corrected
It would be good to show at least one photo of a Petri dish with an inhibition zone.
Answer:
We have plenty of photos for this; we simply presented the inhibition results in a table. I'll show you an example of Enterobacter Ludwigii's resistance to antibiotics:
As an example, Figure 9 shows the results of E. ludwigii for antibiotic resistance (FOX, LVX, CIP, OFX, KAN, GMN, AMK, CMN, ERY, TET, QDA).
Figure 9: Antibiotic susceptibility of Enterobactere ludwigi isolates. 1. ERY (erythromycin) , 2. GMN (gentamycin), 3. AMK (amikacin) , 4. QDA (quinupristin/dalfopristin), ,5. FOX (cefoxitin) , 6. TET (tetracyclin) ,7. LVX (levofloxacin), 8. OFX (ofloxacine) ,9. KAN (kanamycin),10. CMN (clindamycin) ,11. CIP (ciprofloxacin) ,12.RA (Rifampicine).
Materials and methods
440 (please add a mass of extract) Here, definitely a mass of extract is required.
Answer:
The extract yield was 16%. The mass was 16g per 100g of plant material.
Acute oral toxicity study in mice
The same comment as in the results, since information can be added to either section: how were symptoms of toxicity or adverse effects monitored?
Answer:
This was answered in the other section.
4.4.1. Collection of urine samples
In this section, a more detailed description of patients is required. It should be stated how many male and female patients, and how many patients are in each group (15-75 years is a very broad interval). Maybe it makes sense to introduce a small table showing all this data.
Answer:
442 patient samples were collected, as shown in the table:
Table 7: Distribution of the 442 samples according to age and sex
|
Age (Years) |
Female |
Male |
|
15 - 24 |
32 |
70 |
|
25 - 45 |
57 |
80 |
|
46 - 75 |
68 |
135 |
Conclusions
It would be good to name at least the most abundant compounds of the plant identified by HPLC.
Answer:
The most important compound in the extract is epicatchin (47.516 mg/g), followed by catchin (3.208 mg/g), luteolin (1.54 mg/g), rutin (1.472 mg/g), caffeic acid (1.392 mg/g), and apigenin-7-glucoside (1.272 mg/g).

Round 2
Reviewer 1 Report
Comments and Suggestions for Authors
Thank you for the opportunity to review your manuscript. The topic is timely and of significant relevance given the global burden of multidrug-resistant (MDR) infections and the urgent need for new therapeutic alternatives. The study demonstrates a rigorous approach in evaluating the antimicrobial properties of Haloxylon scoparium methanolic extract against clinically isolated MDR bacteria.
Overall, the manuscript has been substantially improved. All reviewer comments appear to have been adequately addressed:
-
The mass of the extract has been clearly stated.
-
A summary table (Table 1) has been provided to present sequencing alignment data more clearly.
-
Statistical analysis, including one-way ANOVA, has been described to validate dose-response findings.
-
The conclusion has been appropriately expanded to suggest future directions, including mechanistic studies.
Final note: While ANOVA was mentioned, consider adding a brief note on whether assumptions for ANOVA were tested (e.g., normality, homogeneity of variances).
Comments on the Quality of English LanguageThe manuscript would benefit from careful English language editing by a native speaker or professional service to enhance readability and ensure clarity.
Author Response
Dear Reviewer,
Thank you very much for reviewing this manuscript. We have improved according to the suggestion.
Expert 1: round 2
Final note: While ANOVA was mentioned, consider adding a brief note on whether assumptions for ANOVA were tested (e.g., normality, homogeneity of variances).
Answer
I have added a brief note in the ‘Methodology’ section discussing the assumptions of ANOVA, specifically normality and homogeneity of variances.
Methodology
A one-way ANOVA was applied to compare the mean inhibition zones between the different inhibitory zones.
concentrations. The hypotheses tested are:
- H0: The means are equal across all concentrations.
- H1: At least one concentration has a significantly different mean.
Before conducting the ANOVA, its assumptions were evaluated:
- Normality: The residuals appeared approximately normally distributed based on
visual inspection.
- Homogeneity of variances: Variance across groups was roughly similar, satisfying
the homoscedasticity assumption.